# CONTEXT-LEVEL LANGUAGE MODELING BY LEARNING PREDICTIVE CONTEXT EMBEDDINGS

## ABSTRACT

Next-token prediction (NTP) is the cornerstone of modern large language models (LLMs) pretraining, driving their unprecedented capabilities in text generation, reasoning, and instruction following. However, the token-level prediction limits the model's capacity to capture higher-level semantic structures and long-range contextual relationships. To overcome this limitation, we introduce **ContextLM**, a framework that augments standard pretraining with an inherent **next-context prediction** objective. This mechanism trains the model to learn predictive representations of multi-token contexts, leveraging error signals derived from future token chunks. Crucially, ContextLM achieves this enhancement while remaining fully compatible with the standard autoregressive, token-by-token evaluation paradigm (e.g., perplexity). Extensive experiments on the GPT2 and Pythia model families, scaled up to 1.5B parameters, show that ContextLM delivers consistent improvements in both perplexity and downstream task performance. Our analysis indicates that next-context prediction provides a scalable and efficient pathway to stronger language modeling, yielding better long-range coherence and more effective attention allocation with minimal computational overhead.

## 1 INTRODUCTION

Large language models (LLMs) have emerged as the foundation of modern natural language processing (NLP), demonstrating remarkable capabilities in text generation, logical reasoning, and generalization ability (Radford et al., 2019; Brown et al., 2020; Achiam et al., 2023; Touvron et al., 2023). These advances are predominantly driven by the autoregressive next-token prediction (NTP) objective, where models are trained to minimize the negative log likelihood of the next token given its preceding sequence. Building on this well-established NTP training objective, researchers have steadily scaled up model size and data, consistently achieving improvements in both language modeling and downstream tasks performance (Kaplan et al., 2020; Hoffmann et al., 2022).

Nevertheless, the reliance on token-level prediction introduces inherent limitations. NTP primarily enforces local sequential consistency, and struggles to capture higher-level semantic and discourse structures that extend beyond the immediate context (Bachmann & Nagarajan, 2024). The historical progression of language modeling, i.e., from character-level (Graves, 2013) to token-level prediction and from bidirectional (Devlin et al., 2019; Lewis et al., 2020) to unidirectional modeling, underscores a consistent principle: more challenging pretraining tasks often yield more powerful models. In

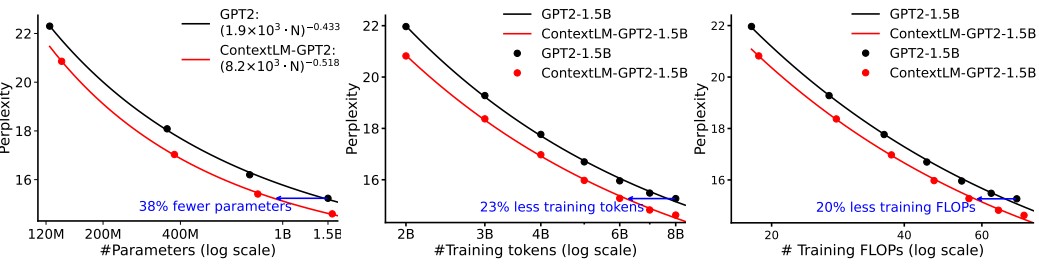

Figure 1: Scaling performance comparison across three dimensions: parameters, training tokens, and training FLOPs for GPT2 and ContextLM-GPT2.

response, some recent works (Gloeckle et al., 2024; Jiralerspong et al., 2023) proposed to go beyond NTP by simply predicting multiple tokens into the future, resulting in Multi-token prediction (MTP).

Moving beyond such token-level extensions requires fundamental architectural innovations. A key challenge is how to construct and leverage higher-level representations. To this end, Joint Embedding Predictive Architectures (JEPA) (LeCun, 2022) proposed to construct and predict a higher level of abstraction in latent space, LCM et al. (2024) leverages a predefined encoder to construct sentence-level representations for token sequences, and Ye et al. (2023) learns global representation by restricting attention spans. A distinct line of research introduces hierarchy at a lower level of granularity by grouping characters into patches. For instance, BLT (Pagnoni et al., 2024) uses a learned encoder with an entropy-based threshold to form these patches, while H-Net Hwang et al. (2025) enables dynamic patching during training via an intermediate smoothing stage.

In this work, we propose **ContextLM**, a framework designed to learn predictive, context-level representations within a hierarchical architecture while remaining fully compatible with the standard token-level paradigm. The core of our model is a Context Predictor, a latent module trained to (i) encode preceding tokens into context embeddings, (ii) autoregressively forecast the embedding of the next context, and (iii) fuse this predictive context embedding back into the token-level decoding process. Crucially, the context predictor receives error signals aggregated from multiple future tokens, enabling the model to inherently learn abstractions that extend beyond immediate local dependencies. This allows ContextLM to jointly model fine-grained lexical patterns and higher-level contextual structures, leading to improved coherence and semantic consistency. A key advantage of our design is its minimal intrusion into the standard Transformer backbone, enabling seamless integration with existing LLMs and evaluation using mainstream metrics like token-level perplexity.

Our empirical evaluation demonstrates that ContextLM achieves superior performance across three critical scaling dimensions: parameters, training tokens, and training FLOPs, as shown in Figure 1. On both GPT2 (Radford et al., 2019) and Pythia (Biderman et al., 2023) backbones, it consistently reduces perplexity relative to baselines. Moreover, it achieves better downstream task performance and exhibits stronger instruction-following abilities after fine-tuning. Comprehensive ablation studies on context chunk size and decoder depth further confirm that these improvements are both stable and robust. Extensive experiments and findings validate that ContextLM is a principled step toward unifying fine-grained token-level modeling with higher-level semantic abstraction, opening a promising direction for the next generation of language models.

## 2 ContextLM

In this section, we present ContextLM, which integrates context embedding prediction with conventional next-token modeling. We first introduce the preliminary and problem setup, then describe the model architecture and training objective of ContextLM. In addition, we provide an analysis of the computational complexity in Appendix B.1.

### 2.1 Preliminary and Problem Setup

**Preliminary.** Given a text corpus $\mathcal{C}$, the standard language modeling objective is to model sequence $\mathbf{x}_{0:T-1} = (x_0, x_1, \ldots, x_{T-1})$, where $\mathbf{x}_{0:T-1} \in \mathcal{C}$, using a unidirectional autoregressive model $\pi_0$. The model $\pi_0$ is typically trained with the teacher-forcing objective, minimizing the negative log-likelihood of the ground-truth next tokens. This approach produces a next-token level predictor that estimates $p_{\pi_0}(x_t \mid x_{<t})$ based solely on preceding tokens, without any explicit abstraction of high-level semantic context.

**Problem Setup.** We aim to enhance the base model $\pi_0$ by incorporating high-level contextual semantics that extend beyond dependencies on preceding tokens. Specifically, our objective is to incorporate the model with a sequence of context embeddings $\mathbf{c}$ derived from the low-level token stream. The context predictor module is trained using teacher-forcing to model these embeddings, thereby predicting the context embedding at each step. The core challenge thus becomes constructing a context-aware language model $\pi$ that integrates both the token embedding and the predicted context embedding to facilitate a context-aware NTP: $\pi(x_t \mid x_{<t}, \hat{c}_k)$, where $k = \lfloor t/w \rfloor + 1$ and $w$ denotes the context chunk size. Here, $\hat{c}_k$ is the predicted context embedding corresponding to the current chunk.

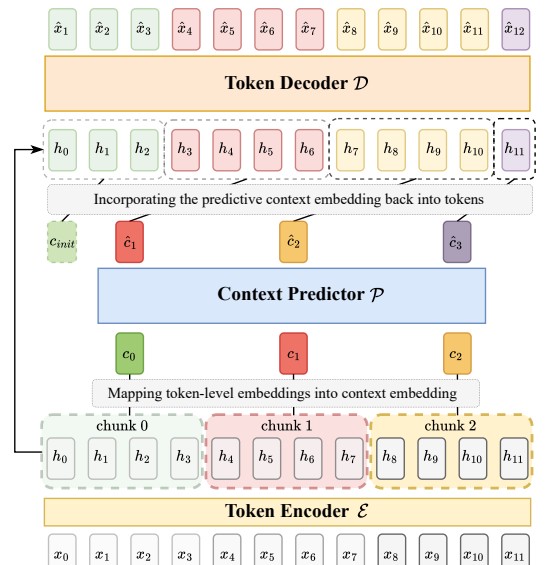

Figure 2: Overview of the ContextLM architecture. The model extends a standard NTP-based backbone by introducing a context predictor $\mathcal{P}$ which generates context embeddings that are fed into the token decoder $\mathcal{D}$ to guide token generation. The pseudocode details the inference procedure, showing the integration of context prediction into the token generation loop.

## 2.2 CONTEXTLM ARCHITECTURE

The overall architecture of ContextLM, illustrated in Figure 2, extends conventional NTP with a context prediction pathway to capture multi-granularity semantic dependencies. It consists of three principal components:

- **Token Encoder** $\mathcal{E}$. Given an input token sequence $\mathbf{x}_{0:T-1}$, the Token Encoder $\mathcal{E}$ encodes it into token-level hiddenstates $\mathbf{h}_{0:T-1} = \mathcal{E}(x_{0:T-1})$. These embeddings serve two purposes: (1) providing fine-grained token representations for token prediction, and (2) supplying the foundation for constructing higher-level context embeddings.

- **Context Predictor** $\mathcal{P}$. We employ a mapping function $f(\cdot)$ to map token embedding $\mathbf{h}_{0:T-1}$ into context embedding $\mathbf{c}_{0:K-1}$, where $K = \lfloor T/w \rfloor + 1$, denoted by $\mathbf{c}_{0:K-1} = f(\mathbf{h}_{0:T-1})$. The Context Predictor $\mathcal{P}$ then performs autoregressive prediction of future context representations, noted by $\hat{\mathbf{c}}_{1:K} = \mathcal{P}(\mathbf{c}_{0:K-1})$. Since the first chunk lacks historical context for the prediction, we introduce a simple placeholder embedding $c_{init}$, forming the complete predicted context sequence $\hat{\mathbf{c}} = (c_{init}, \hat{c}_1, \ldots, \hat{c}_{K-1})$.

- **Token Decoder** $\mathcal{D}$. The Token Decoder $\mathcal{D}$ performs NTP by integrating multi-level information through element-wise addition. Formally, $\mathbf{h} \oplus \hat{\mathbf{c}}^b = (h_0, \ldots, h_{T-1}) \oplus \big(\underbrace{c_{init}, \ldots, c_{init}}_{w-1 \text{ times}}, \underbrace{\hat{c}_1, \ldots, \hat{c}_1}_{w \text{ times}}, \ldots, \underbrace{\hat{c}_{K-1}, \ldots, \hat{c}_{K-1}}_{T+1-(K-1)w \text{ times}}\big)$, where $\hat{\mathbf{c}}^b$ denotes the element wise broadcast $\hat{\mathbf{c}}$, to align with the corresponding token embeddings. Specifically, to preserve the natural NTP paradigm, $\hat{c}_0$ is broadcast only $w-1$ times, while $\hat{c}_{K-1}$ is broadcast additional times to ensure length alignment.

All three components are implemented with causal attention to ensure autoregressive training. Furthermore, even with a one-token left shift, the model is strictly constrained to use only past tokens and contexts at each prediction step, preventing data leakage during multi-level interactions.

## 2.3 TRAINING OBJECTIVE

**The standard NTP** object minimizes the negative log likelihood of the next token, optimized through a token-level error signal solely determined by the preceding sequence:

$$\mathcal{L}_{CE} = -\sum_{t=0}^{T-1} \log \pi_0(x_t \mid x_{<t}),$$

with error signal at time step $t$ given by

$$\frac{\partial \mathcal{L}_{CE}}{\partial h_t} = \frac{\partial \mathcal{L}_{CE}}{\partial z_t} \frac{\partial z_t}{\partial h_t},$$

where $z_t = \mathcal{D}(h_t)$ are the decoder logits. This purely local supervision means that each hidden state $h_t$ only receives feedback from the prediction of its own token.

In **ContextLM**, we retain the standard cross-entropy loss but introduce a key modification by context modeling. The model predicts conditional distributions $\pi_\theta(x_t \mid x_{<t}, \hat{c}_k)$ with decoder logits $z'_t = \mathcal{D}(h_t \oplus \hat{c}_k)$. While the error signals for Token Decoder are identical to conventional Transformers, they become different when back-propagated into the other two components.

- **Context Predictor.** Each predicted context embedding $\hat{c}_k$ receives a joint supervision signal aggregated across all tokens in its chunk:

$$\frac{\partial \mathcal{L}_{CE}}{\partial \hat{c}_k} = \sum_{j \in \mathcal{J}_k} \frac{\partial \mathcal{L}_{CE}}{\partial z'_j} \frac{\partial z'_j}{\partial \hat{c}_k},$$

where $\mathcal{J}_k$ denotes the set of token positions in chunk $k$. Thus, $\hat{c}_k$ is influenced by the aggregated prediction performance across the entire chunk rather than any single position.

- **Token Encoder.** Each token representation $h_t$ now receives two sources of supervision:

$$\frac{\partial \mathcal{L}_{CE}}{\partial h_t} = \underbrace{\frac{\partial \mathcal{L}_{CE}}{\partial z'_t} \frac{\partial z'_t}{\partial h_t}}_{\text{token-level signal}} + \underbrace{\left( \sum_{j \in \mathcal{J}_k} \frac{\partial \mathcal{L}_{CE}}{\partial z'_j} \frac{\partial z'_j}{\partial \hat{c}_k} \right) \frac{\partial \hat{c}_k}{\partial h_t}}_{\text{context-level signal}},$$

meaning that $h_t$ is influenced both by its own token prediction and by the aggregated feedback transmitted through the context embedding $\hat{c}_k$. Compared to NTP, this introduces multi-token supervision into each token representation while still preserving the original local pathway.

Our method allows each token to receive both (i) its own token-level signal and (ii) aggregated multi-token supervision via the context embedding, effectively capturing long-range semantic dependencies while preserving the standard NTP training objective.

## 3 EXPERIMENTS

We conduct extensive experiments to comprehensively assess the effectiveness of ContextLM across different backbones, dataset scales, and evaluation settings. The experiments cover four main aspects: (i) scaling law analysis on GPT2 and Pythia families to examine parameter and data efficiency under controlled budgets (Sec. 3.2); (ii) comprehensive evaluation on diverse downstream tasks (Sec. 3.3); (iii) assessment of instruction-following capabilities through fine-tuning (Sec. 3.4); and (iv) ablation studies and mechanistic analysis of key design choices (Sec. 3.5).

### 3.1 EXPERIMENTAL SETUP

**Backbone and Training Settings.** We evaluate ContextLM on two widely used Transformer families to demonstrate its architectural compatibility and effectiveness. (i) **GPT2**: models are pretrained from scratch on OpenWebText (Gokaslan et al., 2019) with a maximum sequence length of 1024 tokens. (ii) **Pythia**: to examine large-scale scalability, we train ContextLM-Pythia on the Pile dataset (Gao

et al., 2020) (300B tokens) with a maximum sequence length of 2048 tokens, following official hyperparameters including tokenizer, optimizer, learning rate schedule, and batch size.

**Specific Setting for ContextLM.** In our practice, we use the mean pooling function as the mapping function $f(\cdot)$ in Sec. 2.2. We use the first token embedding $h_0$ as $c_{init}$. For Pythia models that use PoSE as the position embedding technique, we use the position of the first token in each chunk as the context chunk position in context layers. Moreover, unless specifically noted, we set the context chunk size to $w = 4$ and use a 2-layer context predictor.

## 3.2 SCALING EXPERIMENTS

### 3.2.1 SCALING LAW ON GPT2

We begin by evaluating the scaling behavior of ContextLM on the GPT2 family, referred to ContextLM-GPT2, trained on OpenWebText under matched compute budgets of the baseline GPT2. Figure 1 presents the perplexity as a function of model parameters, training tokens, and training FLOPs. Across all three scaling dimensions, our model consistently outperforms the GPT2 baseline.

When scaling by parameter count from 124M to 1.5B (Figure 1, left), ContextLM-GPT2 achieves stable perplexity gains over the baseline, indicating that the improvement is preserved as model capacity increases. With a 1.5B model size and increasing training tokens (Figure 1, middle), ContextLM-GPT2 reaches lower perplexity while using 23% fewer training tokens, demonstrating improved data efficiency. Controlling for training FLOPs (Figure 1, right), ContextLM-GPT2 consistently achieves lower perplexity at matched budgets, using 20% less training FLOPs, highlighting that its gains stem from more efficient utilization of compute rather than higher cost.

In addition, details of the experimental setup and downstream task evaluation on GPT2 are provided in the Appendix B.2 and Appendix B.3, respectively. Overall, these results indicate that ContextLM-GPT2 achieves a more favorable scaling law than GPT2: gains are stable with model parameters, convergence requires fewer training tokens, and the performance-to-training FLOPs is consistently improved. This suggests that context-level supervision introduces a richer training signal that scales more efficiently than standard next-token prediction.

Table 1: Perplexity Comparison of Pythia and ContextLM-Pythia across four benchmark datasets. For brevity, we denote ContextLM-Pythia as ContextLM. "Avg PPL" reports the average perplexity per model, where lower values indicate better performance.

| Model | Pile | Wikitext | Lambda OpenAI | Lambda Standard | Avg PPL $\downarrow$ |
|---|---|---|---|---|---|
| Pythia-70M | 18.27 | 57.01 | 142.01 | 973.59 | 297.72 |
| **ContextLM-70M** | **14.96**$_{+3.31}$ | **43.64**$_{+13.37}$ | **71.45**$_{+70.56}$ | **440.77**$_{+532.82}$ | **142.71**$_{+155.01}$ |
| Pythia-160M | 12.56 | 33.44 | 38.20 | 187.28 | 67.87 |
| **ContextLM-160M** | **11.14**$_{+1.42}$ | **28.18**$_{+5.26}$ | **25.97**$_{+12.23}$ | **107.05**$_{+80.23}$ | **43.09**$_{+24.78}$ |
| Pythia-410M | 8.88 | 20.11 | 10.85 | 31.53 | 17.84 |
| **ContextLM-410M** | **8.67**$_{+0.21}$ | **19.50**$_{+0.61}$ | **10.15**$_{+0.70}$ | **24.95**$_{+6.58}$ | **15.82**$_{+2.02}$ |
| Pythia-1B | 7.82 | 16.45 | 7.92 | 17.44 | 12.41 |
| **ContextLM-1B** | **7.66**$_{+0.16}$ | **16.09**$_{+0.36}$ | **7.38**$_{+0.54}$ | **15.75**$_{+1.69}$ | **11.72**$_{+0.69}$ |
| Pythia-1.4B | 7.26 | 14.72 | 6.09 | 10.87 | 9.74 |
| **ContextLM-1.4B** | **7.16**$_{+0.10}$ | **14.61**$_{+0.11}$ | **6.06**$_{+0.03}$ | **10.30**$_{+0.57}$ | **9.54**$_{+0.20}$ |

### 3.2.2 SCALING LAW ON PYTHIA

To examine the general applicability and effectiveness of ContextLM, we further evaluate it on the Pythia family (70M–1.4B parameters) to assess scalability across larger data. As shown in Table 1, ContextLM-Pythia consistently lowers perplexity across the Pile, Wikitext (Merity et al., 2016), and Lambada (Paperno et al., 2016). Our method gains significant improvement across all the test sets and model sizes: at 70M parameters, average perplexity drops from 297.7 to 142.7, a relative

Table 2: Zero-shot and five-shot evaluation results across nine downstream benchmarks. We refer to ContextLM-Pythia as ContextLM for simplicity. "Avg Acc" indicates the mean accuracy per model under each evaluation condition, where higher values indicate better performance.

| Model | Lambada OpenAI | ARC-E | Lambada Standard | ARC-C | Wino Grande | PIQA | Hella-Swag | SciQ | RACE | Avg Acc ↑ |
|---|---|---|---|---|---|---|---|---|---|---|
| **0-shot** | | | | | | | | | | |
| Pythia-70M | 18.3 | 36.9 | 13.4 | 18.5 | 52.1 | 60.0 | 26.6 | 60.5 | 24.9 | 34.6 |
| **ContextLM-70M** | 28.0 | 40.7 | 17.2 | 18.6 | 52.3 | 60.5 | 27.5 | 72.1 | 27.0 | **38.2**+3.6 |
| Pythia-160M | 32.7 | 43.8 | 21.5 | 19.5 | 53.4 | 61.5 | 28.5 | 74.3 | 27.9 | 40.3 |
| **ContextLM-160M** | 37.2 | 45.0 | 25.6 | 19.5 | 52.0 | 62.9 | 29.2 | 77.6 | 28.7 | **42.0**+1.7 |
| Pythia-410M | 51.6 | 52.2 | 36.4 | 21.3 | 53.9 | 66.8 | 33.8 | 81.2 | 30.7 | 47.5 |
| **ContextLM-410M** | 52.2 | 51.7 | 39.0 | 22.7 | 52.3 | 67.4 | 34.3 | 83.0 | 30.8 | **48.2**+0.7 |
| Pythia-1B | 55.9 | 56.8 | 42.0 | 24.2 | 52.5 | 70.5 | 37.7 | 83.3 | 32.7 | 50.6 |
| **ContextLM-1B** | 57.8 | 55.3 | 43.7 | 25.9 | 55.0 | 70.1 | 37.6 | 86.1 | 32.1 | **51.5**+0.9 |
| Pythia-1.4B | 61.6 | 60.4 | 49.7 | 25.9 | 57.5 | 70.8 | 40.4 | 86.4 | 34.1 | 54.1 |
| **ContextLM-1.4B** | 61.6 | 58.9 | 51.4 | 27.2 | 55.7 | 71.2 | 40.6 | 87.9 | 34.9 | **54.4**+0.3 |
| **5-shot** | | | | | | | | | | |
| Pythia-70M | 11.9 | 36.7 | 9.2 | 17.1 | 50.5 | 58.7 | 26.7 | 57.8 | 25.1 | 32.6 |
| **ContextLM-70M** | 19.2 | 41.4 | 14.2 | 18.5 | 51.1 | 60.8 | 27.7 | 71.5 | 25.7 | **36.7**+4.1 |
| Pythia-160M | 24.9 | 44.7 | 19.0 | 18.4 | 50.4 | 63.5 | 28.6 | 76.4 | 27.8 | 39.3 |
| **ContextLM-160M** | 29.1 | 45.7 | 23.1 | 19.8 | 51.9 | 63.1 | 29.5 | 81.7 | 28.2 | **41.3**+2.0 |
| Pythia-410M | 43.9 | 54.7 | 32.8 | 22.3 | 53.4 | 68.0 | 33.8 | 88.9 | 30.4 | 47.6 |
| **ContextLM-410M** | 44.6 | 54.8 | 34.9 | 23.0 | 52.7 | 68.0 | 34.3 | 89.5 | 30.9 | **48.1**+0.5 |
| Pythia-1B | 48.3 | 58.6 | 35.8 | 25.4 | 52.8 | 71.3 | 37.7 | 91.6 | 31.7 | 50.4 |
| **ContextLM-1B** | 49.9 | 60.2 | 39.5 | 24.2 | 54.9 | 70.5 | 38.1 | 91.5 | 32.6 | **51.3**+0.9 |
| Pythia-1.4B | 54.5 | 63.1 | 44.5 | 28.8 | 57.1 | 71.0 | 40.5 | 92.4 | 34.6 | 54.1 |
| **ContextLM-1.4B** | 55.7 | 62.3 | 46.7 | 28.5 | 56.8 | 72.4 | 41.1 | 93.3 | 35.0 | **54.6**+0.5 |

improvement exceeding 50%. At larger scales, the gap remains stable, with ContextLM-Pythia-1B reaching 11.72 versus 12.41 for Pythia-1B, and ContextLM-Pythia-1.4B achieving 9.54 versus 9.74. The consistent improvement in the scale of all models demonstrates that ContextLM is also highly scalable and computationally efficient on large-scale datasets.

## 3.3 DOWNSTREAM TASK EVALUATION

We evaluate the zero-shot and five-shot performance using the `lm-evaluation-harness` [1] across nine benchmarks, which we categorize into three core capabilities: linguistic understanding (Lambada OpenAI/Standard (Paperno et al., 2016), WinoGrande (Sakaguchi et al., 2019)), commonsense reasoning (ARC-Easy/Challenge (Clark et al., 2018), PIQA (Bisk et al., 2019), HellaSwag (Zellers et al., 2019)), and complex reasoning (SIQA (Liu et al., 2025), RACE (Lai et al., 2017)), following the evaluation established in (Gu et al., 2024; Zeng et al., 2025). As shown in Table 5, ContextLM-Pythia consistently outperforms the Pythia baseline across all model sizes under both evaluation settings.

---

[1] https://github.com/EleutherAI/lm-evaluation-harness

The model achieves approximately 10% relative gain on reasoning tasks at smaller scales and maintains a stable 1-2% absolute advantage in larger models. Significant improvements are observed across linguistic understanding (particularly contextual prediction), commonsense reasoning (5-8% gain in physical inference), and complex reasoning tasks (3-5% higher accuracy in multi-step reasoning). These enhancements are especially pronounced in few-shot settings, confirming that context-level supervision enables more effective parameter utilization and stronger semantic understanding throughout the scaling trajectory.

### 3.4 INSTRUCTION-FOLLOWING ABILITY EVALUATION

Finally, we further fine-tune ContextLM-Pythia and Pythia on the Alpaca dataset (Taori et al., 2023), then evaluate their instruction-following capability using MT-Bench (Zheng et al., 2023). As shown in Figure 3, ContextLM-Pythia consistently surpasses Pythia across multiple capability subtasks. At the 1B parameter size, the average score improves from 1.62 to 1.83, while the 1.4B model shows more substantial gains, increasing from 1.99 to 2.37. These results demonstrate that context-level supervision significantly enhances the model's ability to understand and execute instructions.

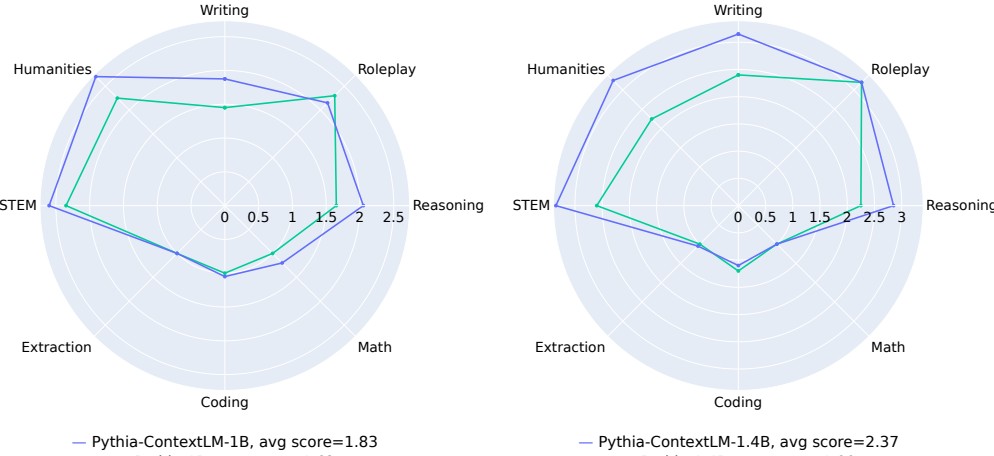

Figure 3: Instruction-following evaluation on MT-Bench across multiple subtasks.

### 3.5 ABLATION AND ANALYSIS

Beyond the main experimental results, we perform additional ablation studies and analyses to investigate how key architectural configurations and modeling strategies influence the behavior and performance of ContextLM. Specifically, we analyze: (i) the impact of chunk size, context predictor depth, and token encoder/decoder depth, (ii) the ability of processing long sequences, and (iii) the visualization of attention distribution patterns.

#### 3.5.1 ABLATION STUDY OF CONTEXTLM ARCHITECTURAL COMPONENTS

**Chunk Size**: We vary the chunk size $w$ among $\{2, 4, 8, 16\}$. As shown in Figure 4 (left), we observe a clear trade-off between performance and computational efficiency. A smaller chunk size (e.g., $w = 2$) yields the best perplexity (20.65), as it provides more frequent and finer-grained contextual guidance to the token decoder. As $w$ increases, the effective sequence length for the context predictor decreases, which reduces its computational and memory overhead but also coarsens the predictive signal, leading to a gradual increase in perplexity (to 21.01 for $w = 16$). Crucially, all configured chunk sizes significantly outperform the vanilla GPT2-base baseline (22.38), demonstrating the robustness of our approach to this design choice.

**Context Predictor Depth**: We next analyze the effect of the context predictor depth (Figure 4, middle). The results show that a lightweight decoder with only 2 layers is sufficient to achieve notable gains. Adding more layers does not yield significant improvement, indicating that the context predictor effectively captures contextual transitions with minimal depth. This efficiency aligns with

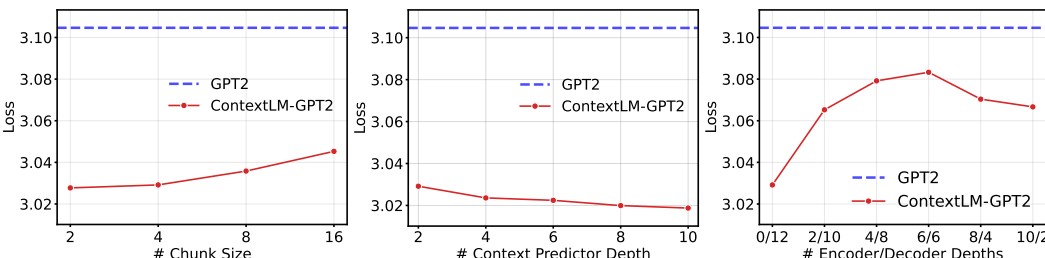

Figure 4: Validation loss on OpenWebText for baseline GPT2 and ContextLM-GPT2 across chunk sizes, context predictor depths, and token encoder/decoder depths.

our design of an auxiliary pathway that enhances representational continuity without introducing unnecessary complexity.

**Model Architecture**: As shown in Figure 4 (right), the 0/12 architecture stands out as the optimal configuration, achieving the lowest perplexity (20.68) due to its exclusive focus on a decoder-only design. This superior performance aligns with the autoregressive nature of language modeling, which primarily depends on a strong decoder for prediction accuracy. In contrast, all the other configurations demonstrate performance degradation (perplexity increasing 3.7% to 5.6%).

### 3.5.2 LENGTH EXTENSION

Modeling long contexts remains challenging for modern LLMs due to the limitations of token-level self-attention, which tends to capture local rather than long-range dependencies as the sequence length increases. ContextLM alleviates this limitation by introducing a context-level pathway that operates on chunked sequences, effectively reducing the sequence length by a factor of $w$. This design enables the model to predict context-level semantics at a coarser granularity, offering guidance that complements token-level attention and alleviates the direct modeling burden of long-range dependencies.

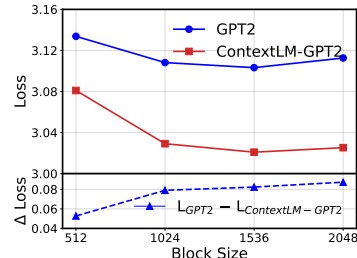

Figure 5: Analysis of long-range modeling ability. The $\Delta Loss$ between ContextLM-GPT2 and GPT2 grows as block size increases.

To assess this capability, we compare the test loss as the sequence length grows from 512 to 2048 tokens. As shown in Figure 4 (right), the $\Delta Loss$ between ContextLM-GPT2 and GPT2 increases with context length, highlighting the stronger benefits of context-level supervision in longer sequences. These results demonstrate that ContextLM not only improves training efficiency, but also enhances coherence and consistency in long-context modeling.

### 3.5.3 ATTENTION DISTRIBUTION VISUALIZATION

To qualitatively understand ContextLM's behavior, we analyze a representative example in Figure 6. ContextLM-GPT2 exhibits significantly concentrated attention allocation compared to the GPT2 baseline model. Specifically, a 59.0% increase in attention to the anaphoric term "*this*" (token id: 18, in Chunk 4), and attention to the key technical descriptors - "*revolutionary*" (token id: 5), "*graphene*" (token id: 16), and "*battery*" (token id: 7) - shows a collective increase of 67.0%. While both models demonstrate limited attention to the explicit reporting clause "analysts said" (token id: 32 33), ContextLM strengthens focus on the analytical context surrounding these tokens, including "*technology*" (token id: 18) and "*analysts*" (token ids: 32-33), with a 16.2% improvement in attention to this analytical span (token ids: 30-34). This pattern indicates ContextLM's enhanced ability to capture both lexical patterns and high-level contextual relationships.

## 4 RELATED WORK

**Hierarchical Architectures:** Several works have explored hierarchical architectures for language modeling. However, these efforts have been largely driven by the goal of computational efficiency rather than performance through explicit modeling of higher-level semantic abstractions. For example, BlockFormer (Ye et al., 2023) processes text in blocks or chunks to capture broader context while

**Input Text:**

Apple unveiled its latest iPhone model on Tuesday. The new device features a revolutionary graphene battery. This technology allows for significantly faster charging times and longer overall battery life, analysts said.

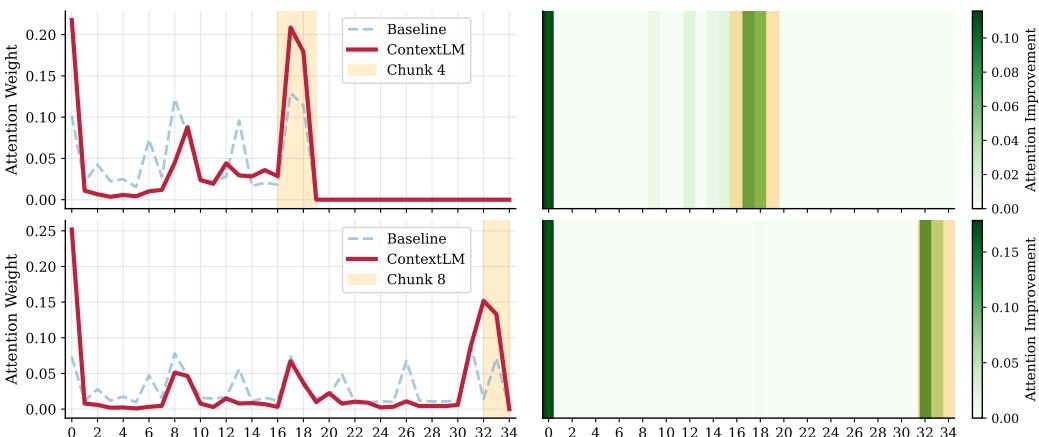

Figure 6: Attention weight analysis for the input text. ContextLM-GPT2-1.5B (orange) shows a significant increase in attention over the baseline GPT2-1.5B (blue) towards the technical concept in Chunk 4 ("*battery. This technology*") and the contextual framing in Chunk 8 ("*analysts said.*"), indicating improved contextual understanding.

improving efficiency. Block Transformer (Luo et al., 2023) adopts a global-to-local modeling strategy to accelerate inference with performance preserved. Similarly, patch-based training methods (Jiralerspong et al., 2023; Wang et al., 2024) have also demonstrated benefits in both efficiency and context length extending rather than improving next-token prediction. Multi-token prediction (MTP) (Gloeckle et al., 2024) shares a similar motivation of predicting beyond the immediate next token, though its focus remains primarily on optimizing the loss function rather than introducing new architectural components. SegFormer (Liu et al., 2023) adaptively determines segment boundaries using linguistic cues, partially aligning with hierarchical modeling. For long-sequence modeling, architectures such as Mamba (Gu et al., 2024) ~~(state space models)~~ and LongNet (Ding et al., 2024) ~~(dilated attention)~~ incorporate hierarchical inductive biases to extend effective context length. HAMburger (Liu & Zhang, 2025) accelerates inference via token smashing but does not explicitly model or predict higher-level semantic representations. However, these methods do not explicitly model or predict representations at varying levels of abstraction.

**Latent Space Context Prediction:** More closely related to our approach are works that leverage in latent representation spaces. LCM et al. (2024) employs diffusion models to predict conceptual semantics, but gained a limited increase in performance and is constrained to a pre-defined semantic space. Several others have explored the extraction and use of contextual semantics in latent spaces (Deng et al., 2023; Hao et al., 2024; Pagnoni et al., 2024; Tack et al., 2025; Su et al., 2025). These methods share with ContextLM the goal of capturing higher-level semantic information, but differ in their technical approaches and primary objectives. ContextLM distinguishes itself by making high-level context prediction an explicit and central objective within the generative process, achieving effective hierarchical modeling through a simple and efficient auxiliary structure that complements rather than replaces next-token prediction.

## 5 CONCLUSION

In this work, we introduced ContextLM, a framework that incorporates standard next-token prediction with context-level supervision while remaining fully compatible with the autoregressive paradigm and requiring no changes to the backbone architecture. By forecasting and integrating predictive context embeddings, ContextLM aligns token-level predictions with higher-level semantic structures at minimal computational cost. Experiments on GPT2 and Pythia families up to 1.5B parameters show consistent improvements in perplexity and downstream performance, with gains persisting across model scales. Further analysis demonstrates that context-level supervision strengthens long-range coherence and attention allocation, highlighting next-context prediction as a scalable and efficient direction for advancing large language models.

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

## A   STATEMENT ON LLM USAGE

Large language models are used during the writing of this paper to polish and refine the prose of certain paragraphs, primarily in the introduction and related work sections, to improve clarity and readability.

## B   APPENDIX

This supplementary material reports additional results and details for ContextLM on GPT-2 backbones. It covers three aspects: computational complexity and memory footprint, training hyperparameters, and downstream evaluation. These results confirm that the improvements in the main text hold across model scales and are not due to hyperparameter tuning or evaluation artifacts.

## B.1  Computational Complexity and Memory Footprint

We compare the computational complexity and memory requirements of ContextLM with those of a vanilla Transformer model with an equivalent parameter budget. The analysis demonstrates that our approach maintains efficiency while enhancing contextual modeling capabilities.

- **Compute Complexity.** The token decoder in ContextLM is identical to that in the vanilla Transformer, with a computational cost (FLOPs) of $O(T^2 d)$, where $T$ is the sequence length and $d$ is the embedding dimension. The context predictor, in contrast, operates on chunked sequences of length $K = \lfloor T/w \rfloor + 1$, resulting in an additional cost of $O(K^2 d) = O((T/w)^2 d)$. For example, with $w = 4$, the additional cost is $O(T^2 d/16)$, which corresponds to only a 6.25% overhead, indicating that context modeling introduces minimal extra computation.
- **Memory Footprint.** The memory required for token embeddings is $O(Td)$. The predicted context embeddings form a sequence of length $K$, contributing $O((T/w)d)$ storage. For $w = 4$, this constitutes 25% of the memory required for token embedding. Since these embeddings are processed by a lightweight decoder (typically two layers), the peak activation memory overhead is minimal. Empirically, we observe less than a 5% increase in GPU memory consumption compared to a parameter-matched vanilla baseline.

Overall, ContextLM incurs minimal additional computational and memory overhead relative to the baseline, establishing context-level prediction as a scalable and efficient approach for enhancing language modeling performance while maintaining full compatibility with existing architectures.

## B.2  Training Hyperparameters on GPT2

In our scaling law experiments, we adopt the configurations detailed in Table 3 for the GPT2 model family. All models are trained on the OpenWebText with a maximum sequence length of 1024 tokens. Optimization is performed using AdamW ($\beta_1 = 0.9$, $\beta_2 = 0.95$) with gradient clipping at 1.0 and linear warmup over the first 1,000 steps. Learning rates are scaled according to model size following established practices.

Table 3: Training hyperparameters for GPT2 family models used in the scaling law experiments.

| Model | $n_{\text{layers}}$ | $d_{\text{head}}$ | $n_{\text{model}}$ | learning rate | batch size | tokens |
|---|---|---|---|---|---|---|
| GPT2-Base | 12 | 12 | 768 | 1e-3 | 0.5M | 9B |
| GPT2-Medium | 24 | 16 | 1024 | 8e-4 | 0.5M | 9B |
| GPT2-Large | 36 | 20 | 1280 | 6e-4 | 0.5M | 9B |
| GPT2-XL | 48 | 25 | 1600 | 4e-4 | 0.5M | 9B |

## B.3  Downstream Task Evaluation on GPT2

Experimental results across multiple benchmarks confirm the effectiveness of ContextLM-GPT2 under diverse evaluation settings. As shown in Table 4, our model achieves consistently lower perplexity than the baseline on OpenWebText, Wikitext, and Lambada datasets. Notably, the XL-scale variant attains an average perplexity of 25.93, corresponding to a 17.8% reduction compared to GPT2-XL, indicating significantly improved modeling of long-range dependencies and contextual coherence.

Further evaluation under zero-shot and five-shot settings, summarized in Table 5, demonstrates systematic improvements across nine representative benchmarks, including Lambada, ARC, Wino-Grande, PIQA, HellaSwag, SciQ, and RACE. These gains are consistent across all model scales, with particularly pronounced improvements on reasoning-intensive tasks such as HellaSwag and PIQA. The stable performance enhancement across both perplexity-based and task-based metrics substantiates that context-level supervision strengthens generalization capability and promotes more robust compositional understanding across datasets and model sizes.

Table 4: Perplexity comparisons between GPT-2 and ContextLM-GPT2 across four benchmark datasets. ContextLM consistently achieves lower perplexity across all model scales.

| Model | OWT | Wikitext | Lambada OpenAI | Lambada Standard | Avg PPL |
|---|---|---|---|---|---|
| GPT2-Base | 22.38 | 45.41 | 74.06 | 301.82 | 110.92 |
| **ContextLM-Base** | **20.68**$_{+1.70}$ | **41.45**$_{+3.96}$ | **55.22**$_{+18.84}$ | **231.41**$_{+70.41}$ | **87.19**$_{+23.73}$ |
| GPT2-Medium | 18.10 | 35.78 | 36.53 | 131.43 | 55.46 |
| **ContextLM-Medium** | **17.03**$_{+1.07}$ | **32.08**$_{+3.70}$ | **28.06**$_{+8.47}$ | **95.06**$_{+36.37}$ | **43.06**$_{+12.40}$ |
| GPT2-Large | 16.22 | 30.92 | 26.68 | 79.27 | 38.27 |
| **ContextLM-Large** | **15.41**$_{+0.81}$ | **28.82**$_{+2.10}$ | **20.89**$_{+5.79}$ | **62.09**$_{+17.18}$ | **31.80**$_{+6.47}$ |
| GPT2-XL | 15.25 | 28.98 | 22.76 | 59.17 | 31.54 |
| **ContextLM-XL** | **14.60**$_{+0.65}$ | **27.05**$_{+1.93}$ | **17.46**$_{+5.30}$ | **44.62**$_{+14.55}$ | **25.93**$_{+5.61}$ |

Table 5: Downstream task accuracy across nine benchmarks for GPT2 and ContextLM-GPT2 under 0-shot and 5-shot settings. ContextLM-GPT2 consistently outperforms GPT2 across all model scales.

| Model | Lambada OpenAI | ARC-E | Lambada Standard | ARC-C | Wino Grande | PIQA | Hella-Swag | SciQ | RACE | Avg Acc |
|---|---|---|---|---|---|---|---|---|---|---|
| | | | | **0-shot** | | | | | | |
| GPT2-Base | 27.0 | 42.1 | 20.1 | 18.2 | 49.6 | 60.3 | 27.5 | 67.5 | 27.5 | 37.8 |
| **ContextLM-Base** | 29.2 | 43.8 | 20.3 | 19.3 | 52.9 | 60.7 | 27.6 | 69.5 | 28.4 | **39.1**$_{+1.3}$ |
| GPT2-Medium | 33.3 | 44.1 | 23.9 | 19.3 | 50.5 | 62.9 | 29.0 | 72.7 | 29.1 | 40.5 |
| **ContextLM-Medium** | 36.8 | 46.7 | 26.7 | 19.5 | 50.9 | 62.9 | 29.8 | 74.2 | 29.3 | **41.9**$_{+1.4}$ |
| GPT2-Large | 36.2 | 45.9 | 26.5 | 20.5 | 48.3 | 63.6 | 30.3 | 73.5 | 28.1 | 41.4 |
| **ContextLM-Large** | 40.7 | 47.9 | 29.9 | 21.1 | 51.3 | 65.7 | 31.2 | 76.6 | 31.0 | **43.9**$_{+2.5}$ |
| GPT2-XL | 38.7 | 48.1 | 29.2 | 20.6 | 50.8 | 65.2 | 31.0 | 77.3 | 29.2 | 43.3 |
| **ContextLM-XL** | 41.9 | 49.9 | 32.3 | 21.4 | 51.9 | 66.1 | 32.3 | 77.7 | 31.2 | **45.0**$_{+1.7}$ |
| | | | | **5-shot** | | | | | | |
| GPT2-Base | 18.3 | 40.8 | 17.0 | 18.6 | 51.9 | 59.8 | 27.3 | 69.0 | 26.8 | 36.6 |
| **ContextLM-Base** | 19.4 | 42.2 | 17.9 | 18.8 | 51.7 | 60.8 | 27.8 | 72.4 | 27.2 | **37.6**$_{+1.0}$ |
| GPT2-Medium | 22.8 | 45.2 | 20.4 | 20.1 | 49.0 | 62.5 | 28.9 | 71.8 | 28.5 | 38.8 |
| **ContextLM-Medium** | 24.0 | 47.3 | 22.0 | 20.1 | 51.1 | 64.2 | 29.8 | 75.3 | 28.4 | **40.2**$_{+1.4}$ |
| GPT2-Large | 25.3 | 46.6 | 23.1 | 20.6 | 51.5 | 65.1 | 30.4 | 79.1 | 27.7 | 41.0 |
| **ContextLM-Large** | 29.7 | 49.7 | 26.4 | 21.1 | 51.5 | 65.2 | 31.2 | 82.0 | 30.7 | **43.1**$_{+2.1}$ |
| GPT2-XL | 28.0 | 49.0 | 24.3 | 21.2 | 50.6 | 65.4 | 30.9 | 81.3 | 28.4 | 42.1 |
| **ContextLM-XL** | 30.7 | 50.6 | 29.3 | 22.2 | 51.0 | 65.9 | 32.2 | 83.6 | 30.8 | **44.0**$_{+1.9}$ |

## B.4 Additional Baseline Comparisons

### B.4.1 Parameter-Matched Comparison

To verify that ContextLM's improvements stem from the Context Predictor's modeling capabilities rather than simply the addition of parameters or depth, we trained an aggressively configured parameter-matched baseline. This baseline is a vanilla GPT2 with two additional Transformer layers (e.g., 14 layers total for the Base scale) to explicitly match the total parameter count of ContextLM (Decoder + 2-layer Predictor).

Table 6: Parameter-Matched Baseline Comparison. ContextLM is compared against a vanilla GPT2 augmented with extra layers to match the total parameter count of ContextLM.

| Model | Baseline | Params-Matched Baseline | ContextLM |
|---|---|---|---|
| GPT2-Base | 22.38 | 21.61 | **20.68** +0.93 |
| GPT2-Medium | 18.10 | 17.87 | **17.03**+0.84 |
| GPT2-Large | 16.22 | 16.18 | **15.41**+0.77 |
| GPT2-XL | 15.25 | 15.14 | **14.60**+0.54 |
| **Avg. PPL** | 17.99 | 17.70 | **16.93**+0.77 |

### B.4.2 Comparison with Multi-Token Prediction

We further evaluate ContextLM against the Multi-Token Prediction (MTP) approach. Table 7 presents the results on the GPT2-XL scale. Both models were trained with identical compute budgets. The results indicate that MTP underperforms the standard NTP baseline in this setting, whereas ContextLM yields superior and consistent gains in both perplexity and downstream zero-shot accuracy.

Table 7: Comparison against Multi-Token Prediction. Evaluation performed at the GPT2-XL scale.

| Model | ppl ($\downarrow$) | Lambda OpenAI | ARC-E | Lambda Standard | ARC-C | Wino Grande | PIQA | Hella-Swag | SciQ | RACE | Avg Acc($\uparrow$) |
|---|---|---|---|---|---|---|---|---|---|---|---|
| GPT2-XL-NTP | 15.25 | 38.7 | 48.1 | 29.2 | 20.6 | 50.8 | 65.2 | 31.0 | 77.3 | 29.2 | 43.3 |
| GPT2-XL-MTP | 16.72 | 36.7 | 49.1 | 25.4 | 19.6 | 49.3 | 64.1 | 30.3 | 77.4 | 30.5 | 42.5 |
| **ContextLM-XL** | **14.60** | **41.9** | **49.9** | **32.3** | **21.6** | **51.9** | **66.1** | **32.3** | **77.7** | **31.2** | **45.0** |

### B.4.3 FLOPs-Matched Ablation

To identify the optimal chunk size $w$ without the confounding factor of varying compute costs, we performed a FLOPs-matched ablation study. In our original analysis, a fixed predictor depth $d$ favored smaller chunks due to higher execution frequency. Here, we adjust $d$ inversely to $w$ to maintain a constant FLOPs budget across configurations.

As shown in Table 8, when compute is normalized, the configuration $\mathbf{w = 4, d = 2}$ emerges as the most efficient design choice, offering the optimal balance between contextual granularity and predictor capacity.

Table 8: FLOPs-Matched Ablation on Chunk Size ($w$) vs. Predictor Depth ($d$). By scaling predictor depth $d$ with chunk size $w$, we ensure a fair comparison under a fixed compute budget.

| Configuration | PPL ($\downarrow$) |
|---|---|
| NTP Baseline | 22.38 |
| ContextLM ($w = 2, d = 4$) | 20.83 |
| **ContextLM** ($w = 4, d = 2$) | **20.68** |
| ContextLM ($w = 8, d = 1$) | 20.94 |

## B.5 EVALUATION ON MODERN INSTRUCTION FOLLOWING AND LONG-CONTEXT BENCHMARKS

To demonstrate the relevance and robustness of ContextLM on modern tasks, we extended our evaluation to include instruction-following and long-context reasoning benchmarks using the Pythia-1.4B backbone.

We evaluated instruction-following capabilities using AlpacaEval 2.0 (Dubois et al., 2024) and long-context performance using LongBench (Bai et al., 2024). For LongBench, we utilized dynamic length extrapolation to handle sequences beyond the model's pre-training limit (2048 tokens).

Table 9 shows that ContextLM achieves a 54.48% win rate against the baseline SFT model. Furthermore, Table 10 demonstrates that ContextLM consistently outperforms the baseline across all LongBench task categories, yielding a +3.11 improvement in overall average accuracy.

Table 9: AlpacaEval 2.0 Win Rate Comparison. Evaluated using Pythia-1.4B-SFT as the reference model. ContextLM shows a significant preference win rate, indicating improved instruction-following ability.

| Model | Win Rate (%) | LC Win Rate (%) |
|---|---|---|
| Pythia-1.4B | 45.52 | 45.99 |
| **ContextLM-Pythia-1.4B** | **54.48** | **54.01** |

Table 10: LongBench Performance Summary (0-4k Subtasks). ContextLM demonstrates superior long-context reasoning capabilities, consistently outperforming the Pythia-1.4B baseline across diverse task categories.

| Model | Single-Doc | Multi-Doc | Summary | Synthetic | Code & Class. | Overall Avg. (↑) |
|---|---|---|---|---|---|---|
| Pythia-1.4B | 13.17 | 4.49 | 13.97 | 2.32 | 33.94 | 20.21 |
| **ContextLM-Pythia-1.4B** | **13.41** | **4.92** | **20.19** | **4.39** | **40.40** | **23.32** |

## B.6 TRAINING CURVE COMPARISON

In this section, we compare the training loss convergence of ContextLM-Pythia against the Pythia baseline. Both models were trained on the large-scale Pile dataset (300B tokens). As Figure 7 demonstrated, ContextLM-Pythia consistently maintains a lower loss compared to the baseline, proving that the context prediction objective introduces a more efficient gradient signal that sustains long-term performance gains.

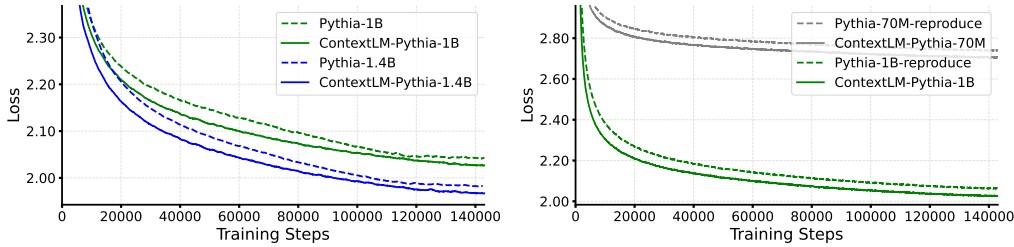

Figure 7: Training curve for Pythia vs ContextLM training on Pile (300B tokens). Left: Comparison against the official Pythia-1B/1.4B training logs. Right: Reproduced 70M and 1B baselines to control for any differences in training configurations.

