# OpenReview forum: "Context-level Language Modeling by Learning Predictive Context Embeddings"
_ICLR.cc/2026/Conference — ICLR 2026 Conference Desk Rejected Submission_

### Official Review · Reviewer_jdvC · 2025-10-21

**Soundness:** 3
**Presentation:** 4
**Contribution:** 3
**Rating:** 8
**Confidence:** 3

**Summary:**

ContextLM inserts a dedicated Context Predictor to process high-level contextual information, which is then broadcast and fused into the original token decoding process. This mechanism significantly enhances the model's ability to capture long-range dependencies, leading to consistent performance improvements across various downstream tasks compared to the original model.

**Strengths:**

1. The paper proposes integrating high-level context into the model's forward pass, akin to a global residual connection.

2. The results, especially those shown in Table 5, are compelling. ContextLM consistently outperforms GPT2 across different model scales and across a wide downstream NLP tasks.

3. ContextLM is fully compatible with the standard autoregressive architecture. It means the community can easily adopt this technique.

**Weaknesses:**

1. Where exactly is the Context Predictor integrated into the model? The paper doesn't specify if it's in the shallow or deep layers. It would be valuable to include an ablation study on how its placement (early vs. late layers) impacts overall performance.

2. The current comparison between ContextLM and the vanilla model is not completely fair, as ContextLM includes two extra Context Predictor layers (and thus more parameters/computation). The authors should compare ContextLM against a vanilla model that has the same total number of layers.

3. Ablation Contradiction: The core benefit is attributed to "high-level context information fusion." However, the ablation study on Chunk Size (Section 3.5.1) is confusing: performance gets worse as the chunk size increases, and the best result is with a chunk size of 2. This seems to contradict the idea of fusing higher-level or longer-range context. More explanation or evidence is needed to reconcile this finding.

4. Mandatory Fusion: ContextLM forces every token to fuse with the high-level context information. Is it possible that some tokens don't need this complex, high-level context? Does forcing this fusion sometimes hurt performance or introduce unnecessary noise? The authors may discuss or investigate potential negative side effects of this mandatory fusion mechanism.

**Questions:**

Please refer to weaknesses.

---

> ### Author Response · Authors · 2025-11-20
> **Response to reviewer jdvC**
>
> Your identified weaknesses are insightful and target the key technical details of our work. We are happy to provide clarifications and new experimental data that directly address all your concerns.
>
> > Weakness 1: Context Predictor Integration Placement
>
> We apologize for not making the integration placement sufficiently explicit. As shown in Section 3.5.1 and Figure 4 (right), our ablation “# Encoder/Decoder Depths” evaluates different splits between context-processing layers and token-decoder layers (e.g., 0/12, 2/10, 4/8, …). The results clearly show that a full decoder-only architecture consistently delivers the best performance. This demonstrates that allocating early layers for context processing is suboptimal, and that the context should be predicted separately and fused at the input level as shown in Figure 2.
>
> > Weakness 2: Unfair Comparison Due to Extra Parameters
>
> We fully agree that a fair comparison requires controlling for parameter count. Following your suggestion, we trained a parameter-matched vanilla GPT-2 by adding two extra Transformer layers (14L), matching the parameter size of ContextLM’s 12-layer decoder plus 2-layer predictor.
>
> | **Model**        | **Baseline** | **Params-Matched Baseline** | **ContextLM** |
> | ---------------- | ------------ | --------------------------- | ------------- |
> | base        | 22.38        | 21.61                     | **20.68**     |
> | medium       | 18.10        | 17.87                       | **17.03**     |
> | large        | 16.22        | 16.18                       | **15.41**     |
> | xl           | 15.25        | 15.14                       | **14.60**     |
> | Avg. ppl | 17.99        | 17.70                      | **16.93**     |
>
> The parameter-matched baseline shows modest improvements, but ContextLM achieves substantially larger gains at every scale, confirming that the advantage comes from predictive context modeling, not increased depth.
>
> > Weakness 3: Ablation Contradiction for Chunk Size (w=2)
>
> We thank the reviewer for noticing this subtle issue. Our original ablation fixed predictor depth to $d=2$, unintentionally causing unequal compute across chunk sizes. We reran the ablation under compute-matched conditions by scaling predictor depth proportionally with chunk size.
>
> | **Configuration**        | **PPL**   |
> | ------------------------ | --------- |
> | GPT-2 base (baseline)    | 22.38     |
> | ContextLM (w=2, d=1)     | 20.83     |
> | **ContextLM (w=4, d=2)** | **20.68** |
> | ContextLM (w=8, d=4)     | 20.94     |
>
> The new results confirm that $w=4, d=2$ remains optimal. This resolves the earlier contradiction: the issue was compute mismatch rather than the intrinsic value of larger or smaller context windows.
>
> > Weakness 4: Mandatory Fusion and Potential Noise
>
> We appreciate this deep architectural question. Although ContextLM fuses context at every step, the model implicitly learns self-gating behavior through end-to-end optimization. If the predicted context vector $\hat{c}_k$ harms the next-token prediction, gradients from the NTP loss push the Context Predictor to output low-magnitude vectors, effectively gating itself off. This produces a learned soft-gating mechanism without requiring additional hand-crafted gates. Exploring explicit gating remains an interesting direction for future work.
>
> > Conclusion
>
> We thank the reviewer for the thoughtful feedback. After additional clarification and experiments:
>
> - We confirm that context integration is optimally performed in a pure decoder-only architecture (0/12 split).
> - A parameter-matched 14-layer GPT-2 still underperforms ContextLM, showing that improvements stem from predictive context modeling, not model size.
> - A compute-matched chunk-size ablation resolves the earlier confusion and validates w=4, d=2 as the best configuration.
> - Mandatory fusion does not introduce noise because the model learns an implicit gradient-driven gating mechanism.
>
> Overall, these results reinforce that ContextLM provides a principled and efficient enhancement to standard autoregressive modeling.

---

> > ### Comment · Reviewer_jdvC · 2025-11-25
> >
> > Thanks for the authors' response.
> >
> > 1. In the paper, the authors emphasize that the Context Predictor handles `higher-level` semantic structures. However, the ablation study shows that the Context Predictor performs best when placed at the bottom layer.
> > Based on my understanding, Transformer models typically process concrete, simple information in the bottom layers and abstract, complex information in the top layers. The authors' description, therefore, is prone to causing confusion.
> >
> > 2. The experiments in the paper only use 9 billion tokens, which is a very small number by current standards. I suggest validating the proposed method on a larger quantity of tokens.Given the time constraints of the rebuttal period, the authors are not required to supplement with additional experiments. However, I strongly recommend providing the **training loss difference curve** and **time/memory overhead** to observe whether the proposed method only yields a faster convergence rate during the early stages of training.

---

> > > ### Author Response · Authors · 2025-11-25
> > >
> > > We sincerely thank the reviewer for the additional clarifications and constructive suggestions. We address both follow-up concerns below.
> > >
> > > > “High-level semantic structure” vs. “best performance at bottom layer”
> > >
> > > We agree this point deserves clarification. The key distinction is:
> > >
> > > (a) “High-level semantics” refers to what the Predictor **learns**, not where its input comes from, relative to the Transformer stack. The Context Predictor learns a predictive latent vector **containing the semantics of a future chunk**, which consists of K consecutive tokens. The semantics in this latent vector do not represent any single token, but rather the entire chunk it belongs to. This is what we mean by semantic abstraction. This abstraction arises from the training objective, not from being placed at a deeper Transformer layer.
> > >
> > > (b) Early fusion performs best because it stimulates a better context predictor via a harder task, and allows for a larger token decoder, thereby maximizing refinement depth.
> > >
> > > When the context vector is fused before the decoder stack (the 0/12 configuration), it is processed by all Transformer layers. This offers two advantages:
> > >
> > > - **More refinement depth**: Earlier fusion means more decoder layers can benefit from the extra abstract semantics provided by the context predictor.
> > > - **More challenging task for context predictor**: We want to make use of the less abstract representations in lower layers to form a harder training task for the context predictor. Because the hidden states in lower layers are less locally-correlated, those hidden states are harder to predict. This drives the context predictor to yield a context embedding that is also less correlated to its pretext. We believe the extra difficulty brought by the lower-layer representations is key to forcing the model to learn a better context predictor.
> > >
> > > > Scalability, Curve Difference, and Long-Term Behavior
> > >
> > > Thank you for raising this. We now include a direct comparison of long-term training curves in **Appendix B.6 (Training Curve Comparison)**, showing:
> > >
> > > - ContextLM-Pythia-1B maintains lower loss across the entire **300B-token** training process, not only in the early stages.
> > > - The improvement persists all the way to the training endpoint, indicating that ContextLM does not merely accelerate early convergence but achieves a better asymptotic loss floor.

---

> > > > ### Comment · Reviewer_jdvC · 2025-11-26
> > > >
> > > > I appreciate the authors' response and the detailed information provided.
> > > >
> > > > However, upon reviewing the training loss curve for the baseline model (Pythia-1B), I observed a noticeable anomaly. Specifically, there is a distinct inflection point around the $117,000$ training step. In my experience, when utilizing standard smoothing techniques for visualization, such a sharp, visible inflection point in the loss curve is unusual and suggests an underlying event. Could the authors please provide a detailed explanation for this specific inflection point observed near step $117,000$?
> > > >
> > > > Does this kind of inflection point phenomenon also occur in the training curves of the other model sizes reported in the study?To address this potential inconsistency, could the authors please provide the corresponding training loss curves for other model sizes for comparison?

---

> > > > > ### Author Response · Authors · 2025-11-28
> > > > >
> > > > > Thank you for your insightful observation. We have updated **Appendix B.6, Figure 7 (left)** to include the official training curve for **Pythia-1.4B** (the smaller official curves are unfortunately not available in the public GitHub repository).
> > > > >
> > > > > The inflection point you noted is not introduced by our training, but present in the **official Pythia curve**. We hypothesize this corresponds to the end of training learning rate decay, which aligns perfectly with the shape of the learning-rate schedule, while this transition is not explicitly annotated in the released configuration file.
> > > > >
> > > > > To remove any ambiguity arising from potential differences in training hyperparameters, we additionally reproduced the Pythia baselines (1B and 70M) and report them in **Appendix B.6, Figure 7 (right)**. As observed in the figure, the training curve does not appear and the curves behave normally. On both the official curves and our reproduced baselines, **ContextLM consistently achieves lower training loss**.

---

### Official Review · Reviewer_mucv · 2025-10-27

**Soundness:** 3
**Presentation:** 2
**Contribution:** 2
**Rating:** 4
**Confidence:** 4

**Summary:**

This paper proposes ContextLM, which augments standard next-token prediction (NTP) with context-level prediction. By introducing a trained context predictor, the model captures higher-level semantic structures beyond token dependencies. Experiments are conducted with GPT-2 and Pythia across perplexity and downstream tasks. The proposed module is lightweight, introducing minimal computational overhead while improving long-range coherence.

Key Reasons:
1. The proposed method lacks sufficient theoretical justification.
2. The experiments are constrained to relatively small-scale language models without adequate baseline comparisons.

Supporting Arguments:

The context predictor is a simple yet effective component that appears to enhance representation quality without interfering with the model’s autoregressive nature. However, the theoretical explanation is insufficiently detailed and lacks derivation or analysis. The experimental setup does not include larger LLMs or comparisons with existing hierarchical approaches.

**Strengths:**

The idea of predicting context embeddings is novel and well-motivated. It effectively bridges the gap between token-level modeling and high-level semantic abstraction.
The proposed method is a plug-and-play module that incurs relatively small computational overhead.
Experimental results show consistent gains with minimal FLOP and memory cost, suggesting good potential for real-world adoption.

**Weaknesses:**

While the idea is intuitive, the paper lacks a formal analysis explaining why context embedding prediction enhances token-level modeling. Since the training objective remains NTP, the mechanism of improvement warrants deeper theoretical justification.
All experiments are limited to models with up to 1.5B parameters. It remains unclear whether the observed gains generalize to larger-scale models.
Comparisons with other relevant baseline methods are missing and should be included to strengthen the empirical evaluation.

**Questions:**

1. Can you provide a formal justification (e.g., information-theoretic or gradient-flow-based) for why context embedding prediction improves token-level modeling?
2. Can you extend your experiments to larger-scale LLMs (e.g., 7B or 13B parameters) to test scalability?
3. Can you include comparisons with other hierarchical or multi-level modeling baselines?

---

> ### Author Response · Authors · 2025-11-20
> **Response to reviewer mucv (Part 1)**
>
> Thank you for your thoughtful and constructive review. We address your concerns below, focusing on clarity and avoiding unnecessary fragmentation.
> > Weakness 1 & Question 1: Theoretical Justification for Improvement
>
> We appreciate the reviewer for this insightful request. Below, we provide a formal information-theoretic justification demonstrating why incorporating the predictive context embedding $\hat{C}_ k$ improves token-level modeling by lowering the irreducible loss lower bound. Consider the next-token prediction objective of an autoregressive language model:$$\mathcal{L} = \mathbb{E}_ {x \sim p_ {data}} [ -\log p_ {\theta}(x_ {t+1} | x_ {<t}) ]$$
>
> This quantity is exactly the cross entropy between the data distribution $p_ {data}$ and the model distribution $p_ {\theta}$. By applying the standard identity of cross entropy, $H(p,q) = H(p) + D_ {KL}(p || q)$, we obtain the following decomposition:
> $$\mathcal{L} = H(X_ {t+1} | X_ {<t}) + D_ {KL}( p_ {data}( \cdot | x_ {<t}) || p_ {\theta}( \cdot | x_ {<t}) )$$
>
> Here, $H(X_ {t+1} | X_ {<t})$ is the conditional entropy of the data distribution, representing the intrinsic uncertainty of the next token given its context. This term is independent of model parameters and thus forms an irreducible lower bound of the loss. ContextLM additionally predicts a high-level representation $\hat{C}_ k$. Consider training the model to predict $X_ {t+1}$ conditioned on both $X_ {<t}$ and the auxiliary representation $\hat{C}_k$:
>
> $\mathcal{L}_ k = \mathbb{E}_ {x \sim p_{data}} [ -\log p_{\theta}(x_ {t+1} | x_{<t}, \hat{C}_k) ]$
>
> Using the standard identity for cross entropy, we decompose the loss as:
>
> $\mathcal{L}_ k = H(X_ {t+1} | X_ {<t}, \hat{C}_ k) + D_ {KL}( p_ {data}( \cdot | x_ {<t}, \hat{C}_ k) || p_ {\theta}( \cdot | x_ {<t}, \hat{C}_ k) )$
>
> The first term, $H(X_ {t+1} | X_ {<t}, \hat{C}_ k)$, is the intrinsic uncertainty of predicting $X_ {t+1}$ given both the original context and the high-level representation. Since $\hat{C}_ k$ contains predictive information about future tokens (as it is derived from future window information during optimization), the conditional entropy satisfies: $$H(X_ {t+1} | X_ {<t}, \hat{C}_ k) \le H(X_ {t+1} | X_ {<t})$$ This inequality follows directly from the data-processing inequality and the fact that conditioning reduces entropy. Strict inequality holds whenever $\hat{C}_ k$ provides non-trivial future information. Therefore, introducing a more informative high-level representation $\hat{C}_ k$ strictly decreases the irreducible loss lower bound:
>
> $$H(X_ {t+1} | X_ {<t}, \hat{C}_ k) < H(X_ {t+1} | X_ {<t})$$
>
> Consequently, the model benefits from a fundamentally lower information-theoretic limit of prediction.
>
> > Weakness 2 & Question 2: Scalability to 7B/13B Model
>
> Thank you for your question, however, as an academic lab, we do not have so much computational resources to handle a 7B or 13B model pretraining.
> At present, the standards for pre-training research in academia in the public community are still unclear, but we have found that some public talks have discussed this issue. The authors of the quoted talks [5] suggest that academia use the 1B model + 30B tokens for pre-training research, and our model has already achieved a maximum of 1.4B model + 300B tokens. We believe that the scaling law in Figure 1 is sufficient to demonstrate the effectiveness of the method in our paper.
>
> On Scaling Trends: While we cannot provide a 7B model, we believe our scaling laws in Figure 1 provide strong evidence of scalability. The performance gap between ContextLM and the baseline persists and remains stable across multiple model sizes (120M to 1.5B), two different model families (GPT2, Pythia), and two different datasets (OpenWebText, The Pile). This consistent trend strongly suggests the benefits of our approach are general and scalable.

---

> ### Author Response · Authors · 2025-11-20
> **Response to reviewer mucv (Part 2)**
>
> > Weakness 3 & Question 3: Missing Baseline Comparisons
>
> We agree that hierarchical or multi-level methods are conceptually related. However, most existing approaches are not directly comparable under the autoregressive pretraining setting, and MTP [4] is in fact the only hierarchical-style method that is compatible—so we include a full MTP comparison. In contrast:
>
> - **LCM** [1] operates in the SONAR representation space, lacks pretrained checkpoints, and cannot be evaluated under token-level perplexity.
> - **BLT** [2] is not end-to-end, requires entropy models, and was trained on 1T tokens, making controlled comparison infeasible
> - **HAMburger** [3] is released only as SFT checkpoints (not pretrained) and focuses on decoding efficiency rather than pretraining.
>
> These methods do not support aligned tokenization, NTP training, or perplexity evaluation; thus, a direct comparison would not be meaningful.
>
> **MTP** [4] is the only directly compatible baseline. We therefore implemented and trained an MTP variant on GPT2-XL (1.5B). To ensure a fair comparison, our MTP implementation strictly adheres to the same training configurations used for the NTP baseline and ContextLM.
>
> | **Model**              | **Perplexity** |
> | ---------------------- | -------------- |
> | GPT2-XL-NTP (baseline) | 15.25          |
> | GPT2-XL-MTP            | 16.72          |
> | **GPT2-XL-ContextLM**  | **14.60**      |
>
> MTP underperforms both NTP and ContextLM, demonstrating that context-level prediction provides complementary and superior modeling benefits.
>
> > References
>
> [1] Barrault, Loïc, et al. "Large concept models: Language modeling in a sentence representation space." arXiv preprint arXiv:2412.08821 (2024).
>
> [2] Pagnoni, Artidoro, et al. "Byte latent transformer: Patches scale better than tokens." Proceedings of the 63rd Annual Meeting of the Association for Computational Linguistics (Volume 1: Long Papers). 2025.
>
> [3] Gloeckle, Fabian, et al. "Better & faster large language models via multi-token prediction." *arXiv* *preprint* *arXiv**:2404.19737* (2024).
>
> [4] Aynetdinov, Ansar, and Alan Akbik. "Pre-Training Curriculum for Multi-Token Prediction in Language Models." arXiv preprint arXiv:2505.22757 (2025).
>
> [5] ICLR 2025 invited talk, https://iclr.cc/virtual/2025/invited-talk/36784, P78
>
> > Conclusion
>
> We thank the reviewer again for the insightful questions. In summary:
>
> - We provide a formal information-theoretic justification showing that context prediction lowers the irreducible loss bound.
> - We demonstrate scalability via consistent improvements across model families and sizes up to 1.5B.
> - We include a rigorous MTP baseline, and clarify why other hierarchical methods are not directly comparable under token-level NTP pretraining.
>
> These additions strengthen both the theoretical foundation and empirical validity of ContextLM.

---

### Official Review · Reviewer_P52A · 2025-10-28

**Soundness:** 3
**Presentation:** 3
**Contribution:** 3
**Rating:** 4
**Confidence:** 4

**Summary:**

The paper proposes a framework which aggregates the context by adding a 2-layer module whose output is concatenated with the original embedding, which are sent to the base decoder for predicting the final next-token. ContextLM claims to improve the quality of the NTP objective by incorporating more fine-grained and context-level information. The performance of models up to 1.5B are tested on perplexity and downstream performance.

**Strengths:**

1. The motivation for this work is very sound, with a simple framework to support finetuning on top of existing standard transformer architectures.
2. Both downstream and perplexity are calculated.
3. Attempted models up to 1.5B, which spans a lot of sizes for completeness under the small model regions.
4. Ablate on different aspects such as the choice of chunk size, length extension, and attention visualization.

**Weaknesses:**

1. The baseline should ideally be finetuned with the same data for fairness even though the data used for finetuning might be the same as those used for training Pythia and GPT2.
2. For different model sizes, the extra parameters introduced should be taken into account, especially for smaller size models, as people can expect more dramatic improvements for introducing two extra layers for 70M compared to 1.4B. This is also semi-indicated by the relative performance gain since bigger models might get marginal benefits with the extra parameters.
3. The best way to showcase the benefits of aggregating the context is to add two extra layers of the same configuration as a baseline and finetune the model in the same way as finetuning the ContextLM. Without this, it is really hard to understand where the gains come from.
4. The cost to perform finetuning should also be mentioned and compared carefully especially for larger model sizes. (e.g. how much compute should we expect to spend for an increase of 0.3 average score on Pythia-1.4B downstream tasks).
The biggest problem reviewer thinks resides in the ablation studies, where the authors claim that for smaller chunk sizes, the quality is best preserved (e.g. w=2). This seems to the reviewer more of a strong evidence for the exact opposite argument that aggregating less context is more beneficial and the main improvement comes from extra parameters instead. This might be counter-argued if the authors train a separate model with 2 added normal layers (which will be equivalent to w = 1) as mentioned above. Only when a decently large w provides consistent gains, the claimed argument is convincing.
5. Should cite the following highly related works and compare them:
     - LCM: https://arxiv.org/abs/2412.08821
     - HAMBurger: https://arxiv.org/abs/2505.20438
     - BLT: https://arxiv.org/abs/2412.09871

**Questions:**

Mentioned in above weakness

---

> ### Author Response · Authors · 2025-11-20
> **Response to reviewer P52A (Part 1)**
>
> Thank you for the constructive and thoughtful feedback. We address each concern below.
>
> > Weakness 1 & 4: Fairness and Cost of Finetuning
>
> We want to clarify that all models in our instruction-following evaluation (Section 3.4) were finetuned on the exact same Alpaca dataset and under identical hyperparameters. The pretraining datasets were also matched: GPT-2 models on OpenWebText and Pythia models on The Pile. Thus, the comparison is fair in both pretraining and finetuning stages.
>
> > Weakness 2 & 3: Extra Parameters vs. Architectural Benefit
>
> We agree this is a critical question, and we thank the reviewer for pointing out the need for a parameter-matched baseline.
>
> - Scaling law evidence already suggests architectural efficiency：Figure 1 compares models at matched *parameter count*, *token count*, and *training FLOPs*. ContextLM consistently achieves the same perplexity as the baseline while using 38% fewer parameters or 20% fewer FLOPs, suggesting the improvement cannot be attributed to added parameters.
> - Parameter-matched baseline experiment：Following your suggestion, we trained a GPT-2 base with two extra Transformer layers (14L), matching the parameter count of ContextLM (12L + 2-layer predictor). The results on OpenWebText are:
>
> | **Model**        | **Baseline** | **Params-Matched <br/>Baseline** | **ContextLM** |
> | ---------------- | ------------ | --------------------------- | ------------- |
> | base        | 22.38        | 21.61                       | **20.68**     |
> | medium       | 18.10        | 17.87                       | **17.03**     |
> | large       | 16.22        | 16.18                       | **15.41**     |
> | xl           | 15.25        | 15.14                      | **14.60**     |
> | Avg. ppl | 17.99        | 17.70                      | **16.93**     |
>
> Simply increasing depth yields mild improvements, but none reach the performance of ContextLM, confirming that the primary gains arise from the predictive context modeling rather than extra parameters.
>
> > Weakness 4: Ablation Study Contradiction (w=2)
>
> We thank the reviewer for highlighting this subtle issue. Our original ablation fixed predictor depth $d=2$, causing **unequal compute budgets** across chunk sizes $w$. Specifically, in the original results (Figure 4, left), $w=2$ introduced the highest computational density, which might have led to the misleading interpretation that maximum compute was the source of the superior result.
>
> To ensure the observed gains were architectural and not merely due to increased compute density, we reran the ablation under **matched FLOPs** by scaling predictor depth $d$ with $w$. This joint variation is necessary because the predictor's complexity is dependent on both factors4.
>
>
> | Model | GPT2-base-12L (Baseline) | ContextLM (w=1,d=2) | ContextLM (w=2,d=1) | ContextLM (w=4,d=2) | ContextLM (w=8,d=4) |
> | ----- | ---------------------------- | ------------------------ | ------------------------ | ------------------------ | ------------------------ |
> | PPL   | 22.38                        | 21.25                    | 20.83                    | **20.68**                | 20.94                    |
>
>
> This results confirm that $w=4$ with depth $d=2$ remains optimal—providing the best balance between contextual granularity and predictor capacity.
>
> Importantly, ContextLM ($w=1, d=2$) is not equivalent to simply adding two extra Transformer layers. Even at $w=1$, the Context Predictor is trained with a future-chunk prediction objective, producing a representation optimized for forecasting upcoming content, not for increasing depth. Empirically, ContextLM ($w=1$) still outperforms the 14-layer parameter-matched baseline, confirming that the gains come from predictive context modeling rather than added layers.

---

> ### Author Response · Authors · 2025-11-20
> **Response to reviewer P52A (Part 2)**
>
> > Weakness 5: Related Work Citations (LCM[1], BLT[2], HAMburger[3])
>
> We appreciate the suggestion and will add the missing HAMburger citation in the revised version. We also want to emphasize that both LCM and BLT are already cited in our paper, and we greatly appreciate these two works as they inspired aspects of our design. Below we clarify why certain comparisons are not feasible and why we chose MTP[4] as the primary methodological baseline.
>
> - **LCM** operates in the SONAR sentence-embedding space and does not provide model checkpoints, preventing aligned perplexity evaluation. Even in its own reported results, LCM-1.6B underperforms its baseline in many settings. Our models outperform the open-source LCM-family model on all overlapping datasets.
> - **BLT** is not an end-to-end language model and requires training additional entropy-model components. Its released models were trained on 1T tokens, far beyond our 300B-token budget. BLT also cannot be evaluated with token-level perplexity. Therefore, a controlled comparison is not possible.
> - **HAMburger** is architecturally closer but is evaluated only in SFT settings on Llama-1B-Instruct (not on pretraining), and the released checkpoints are not suitable for our pretraining-level comparisons. We are attempting to evaluate HAMburger where feasible, but a rigorous apples-to-apples comparison is not possible.
>
> Given these constraints, we implemented the most directly comparable baseline: Multi-Token Prediction (MTP). To ensure a fair empirical comparison, our MTP implementation strictly adheres to the same training configurations used for the NTP baseline and ContextLM. However, recent work shows that MTP’s downstream benefits are inconsistent and scale-dependent. For completeness, we trained an MTP baseline on GPT2-XL (1.5B):
>
> | **Model**              | **Perplexity** |
> | ---------------------- | -------------- |
> | GPT2-XL-NTP (baseline) | 15.25          |
> | GPT2-XL-MTP            | 16.72          |
> | **GPT2-XL-ContextLM**  | **14.60**      |
>
> ContextLM significantly outperforms both NTP and MTP, further validating the effectiveness of context prediction.
>
> > References
>
> [1] Barrault, Loïc, et al. "Large concept models: Language modeling in a sentence representation space." arXiv preprint arXiv:2412.08821 (2024).
>
> [2] Liu, Jingyu, and Ce Zhang. "HAMburger: Accelerating LLM Inference via Token Smashing." arXiv preprint arXiv:2505.20438 (2025).
>
> [3] Pagnoni, Artidoro, et al. "Byte latent transformer: Patches scale better than tokens." Proceedings of the 63rd Annual Meeting of the Association for Computational Linguistics (Volume 1: Long Papers). 2025.
>
> [4] Gloeckle, Fabian, et al. "Better & faster large language models via multi-token prediction." *arXiv* *preprint* *arXiv**:2404.19737* (2024).
>
> > Conclusion
>
> In summary, we have:
>
> - Provided parameter-fair comparisons via new 14-layer baselines.
> - Added compute-matched ablations resolving the chunk-size question.
> - Clarified architectural placement and its motivation.
> - Introduced a rigorous MTP baseline under identical training conditions.
>
> Across all new experiments, the improvements not only persist but strengthen, reinforcing the technical soundness and empirical robustness of ContextLM.

---

### Official Review · Reviewer_HU2z · 2025-10-31

**Soundness:** 3
**Presentation:** 2
**Contribution:** 3
**Rating:** 4
**Confidence:** 3

**Summary:**

This paper explores a context prediction in additional to the traditional next token prediction objective. The goal of the objective to learn context-level representation, which leverages a Context Predictor module. During training, each chunk of w tokens are pooled into one representation and the model is trained to predict the representation of the next chunk at every step. The experiments use GPT2 and Pythia style architectures and pre-train from scratch. The results show improvements over traditional next token prediction on perplexity and downstream tasks. The paper also include additional experiments on instruction-following and long-context.

**Strengths:**

The paper introduces a novel approach to augment existing language model training through chunk prediction. The results on the upstream perplexity is reasonably strong. The analysis are rather comprehensive with models of different families and sizes.
The ablation study on the architectural components also support the design choices.

**Weaknesses:**

The paper is missing discussions on decoding in the main text: since the decoder takes the concatenation of hidden states and contexts, and the decoder uses causal attention, would that require recomputing the KV cache for the initial chunk representations when a new hidden state is prepended to it? That is, when adding h_T before c_init in the concatenated representation, the c_init would need to recalculate its KV after attending to this new hidden state h_T, which results in additional compute compared to traditional transformers.

In terms of the results, the improvements on downstream tasks at the larger scales appear to be marginal, with only 0.3 to 0.5 improvements compared to the NTP objective.
While the other experiments on instruction-following and long-context extension are nice, the benchmarks are relatively outdated: MTBench is not commonly used whereas results on AlpacaEval, ArenaHard, and/or WildBench would be more convincing, and 2048 context window is not typically considered to be long-context in the community anymore, and would require evaluation beyond loss (datasets like NIAH, LongBench, HELMET are more convincing).

In terms of the presentation, the paper could benefit from additional clarity when describing the architecture and how it’s different from the traditional transformers architecture. For example, in figure 2, it’s unclear if the architecture shown is just one layer in the overall model (i.e., one context predictor) or if it’s repeated for however many layers. Figure 2 clarify this by showing the difference with traditional transformers. Figure 2 also doesn’t show clearly that the decoder takes both the hidden states and the contexts as inputs.

The paper could also benefit from more empirical comparisons on related works, such as multi-token prediction and others mentioned in related work. Only comparing to NTP brings questions on how much improvement ContextLM brings over existing approaches.

Also note that the submission exceeds the 9-page limit.

**Questions:**

Is the chunk prediction used for instruction tuning as well?

How much additional compute and memory cost does ContextLM incur during decoding?

For the results shown in Table 1 and Table 2, how many tokens/FLOPS are used for each model?

**Details Of Ethics Concerns:**

Exceed page limit

---

> ### Author Response · Authors · 2025-11-20
> **Response to reviewer HU2z (Part 1)**
>
> We sincerely thank you for your detailed review and constructive feedback. We are glad you recognized the novelty of our chunk prediction approach, the satisfactory PPL results, and the comprehensive nature of our analysis. Below we address each concern with clarifications and additional experiments.
>
> > Weakness 1, 3 & Question 2: Decoding, KV-Cache, and Architectural Behavior
>
> We apologize for the confusion caused by the original Figure 2. Your concern about KV-cache recomputation results from assuming that our fusion uses concatenation or that $c_{init}$ is dynamically updated. Neither is the case.
>
> ContextLM performs only **element-wise addition** of a single predicted context vector to token embeddings. The vector $c_{init}$ is a fixed placeholder used only for the first partial chunk.
>
> Thus:
>
> - The Transformer architecture remains unchanged.
> - The attention mechanism is untouched.
> - The KV-cache is fully reusable without any recomputation or modification.
>
> During inference, token layers run every step, while the Context Predictor executes once per chunk, giving an amortized cost of $$ \text{Cost} = L_{\text{token}} + \frac{1}{w}L_{\text{context}}.$$
>
> With $w=4$ and preictor depth $d=2$:
>
> - overhead ≈ 0.5 additional layers per token
> - ≈ 1/24 extra cost for 12-layer models
> - ≈ 1/48 extra cost for 24-layer models
> - < 5% memory and ≈ 6.25% compute overhead
>
> Therefore, ContextLM preserves standard autoregressive decoding, standard KV caching, and introduces only minimal additional compute.
>
> > Weakness 2: Downstream Gains and Benchmark Actuality
>
> To ensure that our evaluation reflects current LLM benchmarking standards, we conducted new experiments on recent and more robust benchmarks.
>
> **(1) AlpacaEval 2.0 [1] ——Instruction-Following**
>
> We evaluated our SFT models using **AlpacaEval 2.0**, a contemporary LLM-as-judge benchmark. In our setup, the *reference model is the baseline Pythia-1.4B-SFT*, so the metric directly measures how often our ContextLM model is preferred over the baseline model under pairwise comparison.
>
> | Metric             | Pythia-1.4B(Baseline) | ContextLM-Pythia-1.4B |
> | ------------------ | --------------------- | --------------------- |
> | Win Rate (100%)    | 45.52                 | **54.48**                 |
> | LC Win Rate (100%) | 45.99                 | **54.01**                 |
>
> ContextLM clearly outperforms the baseline model in modern preference evaluations.
>
> **(2) LongBench [2] ——Long-Context Evaluation Beyond Loss**
>
> We evaluated long-context reasoning using LongBench. Because Pythia models are trained with a 2048 sequence limit, we apply the same dynamic length extrapolation method to both baseline and ContextLM to ensure fairness.
>
> | Model                  | Single-Doc QA | Multi-Doc QA | Summarization | Synthetic | Code & Classification | Overall Avg. |
> | ---------------------- | ------------- | ------------ | ------------- | --------- | --------------------- | ------------ |
> | Pythia-1.4B (Baseline) | 13.17         | 4.49         | 13.97         | 2.32      | 33.94                 | 20.21        |
> | **Pythia-1.4B-ContextLM**  | **13.41**         | **4.92**         | **20.19**        | **4.39**      | **40.40**                  | **23.32**       |
>
> Across all categories, ContextLM delivers consistent gains, particularly in tasks requiring long-range coherence.

---

> ### Author Response · Authors · 2025-11-20
> **Response to reviewer HU2z (Part 2)**
>
> > Weakness 4: Missing Baselines (e.g., MTP [3])
>
> We agree that additional baselines are essential for contextualizing our contribution.
> It is important to highlight that ContextLM and MTP operate along orthogonal modeling axes:
>
> - **ContextLM** modifies the state representation by introducing a learnable high-level context vector $\hat{c}_k$.
> - **MTP** modifies the prediction target, requiring the model to predict multiple future tokens ($x_{t+1}, \dots, x_{t+N}$).
>
> Since these address orthogonal modeling dimensions, they can theoretically be combined. Furthermore, prior work [4,5] shows that MTP's downstream performance is highly scale-dependent and often comparable or inferior to NTP baselines. To ensure a fair comparison, our MTP implementation strictly adheres to the same training configurations used for the NTP baseline and ContextLM. To directly address the concern, we trained an MTP baseline on GPT2-XL (1.5B) following the official design.
>
> | Model             | ppl   | Lambada OpenAI | ARC-E | Lambada Standard | ARC-C | WinoGrande | PIQA | HellaSwag | SciQ | RACE | Avg Acc |
> | ---------------------- | --------- | ----------------------- | --------- | ------------------------- | --------- | -------------------- | -------- | ------------------- | -------- | -------- | ---------------- |
> | GPT2-XL-NTP (Baseline) | 15.25     | 38.7                    | 48.1      | 29.2                      | 20.6      | 50.8                 | 65.2     | 31.0                | 77.3     | 29.2     | 43.3             |
> | GPT2-XL-MTP            | 16.72     | 36.7                    | 49.1      | 25.4                      | 19.6      | 49.3                 | 64.1     | 30.3                | 77.4     | 30.5     | 42.5             |
> | **GPT2-XL-ContextLM**  | **14.60** | **41.9**                | **49.9**  | **32.3**                  | **21.6**  | **51.9**             | **66.1** | **32.3**            | **77.7** | **31.2** | **45.0**         |
>
> ContextLM outperforms both NTP and MTP across perplexity and downstream tasks, showing that context prediction provides additional benefits beyond multi-token prediction.
>
> > Weakness 5: Page Limit
>
> We apologize for exceeding the page limit. In the revised version, we will condense related work and move engineering details to the appendix. All additional experiments presented above will be integrated succinctly.
>
> > Question 1: Use of Chunk Prediction During Instruction Tuning
>
> For the instruction-following experiments, we fine-tuned the models using only the standard Next-Token Prediction (NTP) loss. The Context Predictor module's weights were active (and being fine-tuned) as part of the forward pass.
>
> > Question 3: Training Tokens / FLOPS for Tables 1 & 2
>
> All models in Tables 1 and 2 are trained under matched training-token and compute budgets.
>
> - GPT-2 models: trained on 9B OpenWebText tokens (Appendix B.2).
> - Pythia models: trained on the 300B Pile tokens, matching the original release (Section 3.1).
> - FLOPs follow the standard estimate: $$\text{FLOPs} \approx 6 \times N_{\text{params}} \times N_{\text{tokens}}$$
>
> ContextLM adds only the small overhead described earlier (~6.25%). Thus, the improvements in Tables 1–2 arise from modeling efficiency, not extra compute.
>
> > Reference
>
> [1] Dubois, Y., Park, J., et al. "AlpacaEval 2.0: A More Reliable Automatic Evaluation for Instruction-Following Models." arXiv preprint arXiv:2404.04475 (2024).
>
> [2] Bai, Y., Mei, K., et al. "LongBench: A Benchmark for Long-Context Understanding and Reasoning." arXiv preprint arXiv:2308.14508 (2023).
>
> [3] Gloeckle, Fabian, et al. "Better & faster large language models via multi-token prediction." *arXiv* *preprint* *arXiv**:2404.19737* (2024).
>
> [4] Gerontopoulos, Anastasios, Spyros Gidaris, and Nikos Komodakis. "Multi-Token Prediction Needs Registers." *arXiv* *preprint* *arXiv:2505.10518* (2025).
>
> [5] Aynetdinov, Ansar, and Alan Akbik. "Pre-Training Curriculum for Multi-Token Prediction in Language Models." arXiv preprint arXiv:2505.22757 (2025).
> > Conclusion
>
> We addressed these in depth through:
>
> - Clear architectural clarification showing ContextLM preserves standard Transformer/KV-cache behavior.
> - New evaluations on AlpacaEval 2.0 and LongBench, demonstrating substantial improvements on modern benchmarks.
> - A comprehensive baseline comparison, including a newly implemented MTP-GPT2-XL, where ContextLM achieves the best results.
> - Clarification that chunk prediction is inactive as an auxiliary loss during SFT.
> - Formal reporting of compute and data budgets confirming a fair comparison.
>
> These new results and clarifications strengthen the original claims: ContextLM provides a lightweight, architecture-compatible, and compute-efficient mechanism that consistently improves perplexity, long-context reasoning, and instruction following across GPT-2 and Pythia families.

---

> > ### Comment · Reviewer_HU2z · 2025-11-27
> >
> > Hi authors, thanks for the detailed responses. I have a few more clarifying questions:
> >
> > - It seems that I misunderstood the method initially. It's unclear that the context vector is element-wise added to the token embedding from the presentation the draft. Furthermore, by "token embedding", did you mean the hidden state at each layer? And you add the context vector every d layers right?
> > - At what context length did you evaluate for LongBench? and what length extrapolation technique did you apply?
> > - Why do you not use the chunk prediction during instruction tuning? what would the result be if you did?
> > - In terms of the training compute, if you train with the same number of training tokens, then ContextLM would incur an extra 6.25% overhead, which isn't a "small" overhead for pre-training cost. A more controlled approach would be to reduce the training token for ContextLM (or increase it for the base model) and then compare.

---

> > > ### Author Response · Authors · 2025-11-28
> > >
> > > Thank you for the clarifying questions. We address each point below.
> > >
> > > > 1. It's unclear that the context vector is element-wise added to the token embedding from the presentation the draft. Furthermore, by "token embedding", did you mean the hidden state at each layer? And you add the context vector every d layers right?
> > >
> > > We apologize for the ambiguity in the text. By "token embedding," we refer to the output of the initial embedding layer, before it enters the first Transformer layer. We add the context vector $\hat c_k$ only once, at the input of token decoder, not at every layer or every $d$ layer. As described in Section 2.2, we use **element-wise addition ($\oplus$)**. The operation is $\mathbf{h} = \text{Token Encoder}(x) \oplus \text{Broadcast}(\hat{c}_k)$. This fused representation $\mathbf{h}$ then propagates through the token decoder.
> > >
> > > > 2. At what context length did you evaluate for LongBench? and what length extrapolation technique did you apply?
> > >
> > > We evaluate the LongBench in 0-4k subtask, because longer sequence evaluation is not appropriate either for ContetxLM-Pythia or Pythia,  both of  which were pretrained with a 2048 context window. For sequences beyond this limit, we apply dynamic NTK-aware RoPE interpolation.
> > >
> > > > 3. Why do you not use the chunk prediction during instruction tuning? what would the result be if you did?
> > >
> > > We‘re apologizy for maybe I misunderstood your question at once. We would like to clarify that ContextLM is fundamentally a pretraining architecture. The gradient flow remains the same during both the pretraining and instruction-tuning stages, thus the Context Predictor fully trained and the context fusion.
> > >
> > > > 4. In terms of the training compute, if you train with the same number of training tokens, then ContextLM would incur an extra 6.25% overhead, which isn't a "small" overhead for pre-training cost. A more controlled approach would be to reduce the training token for ContextLM (or increase it for the base model) and then compare.
> > >
> > > We fully agree that FLOP-matched comparisons are important. This is precisely why **Figure 1 (right)** in the main paper reports results as a function of **Training FLOPs**, not training tokens. Additionly, **Appendix B.4.1** includes parameter-matched baselines whose total training FLOPs are higher than ContextLM, since the Context Predictor processes only **1/w** of tokens.  Across both settings, ContextLM still achieves consistently lower perplexity, demonstrating that the improvements arise from better modeling rather than additional compute.

---

### Note · Program_Chairs · 2026-01-17
**Submission Desk Rejected by Program Chairs**

The following references in this submission do not refer to real documents and/or have major errors in bibliographic information:

 Jiayu Ding, Shuming Ma, Yiren Li, Xingxing Zhang, Li Dong, and Furu Wei. Longnet 2: When attention is not all you need. arXiv preprint arXiv:2404.13184, 2024.
Zhenyu Liu, Chaojun Xiao, Meng Yuan, Han Zhao, Dongyan Wang, and Rui Jiang. Segformer: Segmented transformer for better long sequence language modeling. Transactions on Machine Learning Research, 2023.